# Dissipation Alters Modes of Information Encoding in Small Quantum Reservoirs near Criticality

**DOI:** 10.3390/e27010088

**Published:** 2025-01-18

**Authors:** Krai Cheamsawat, Thiparat Chotibut

**Affiliations:** Chula Intelligent and Complex Systems Lab, Department of Physics, Faculty of Science, Chulalongkorn University, Bangkok 10330, Thailand; kraicwt@gmail.com

**Keywords:** quantum reservoirs, driven-dissipative dynamics, partial information decomposition, dynamic instability, memory capacity

## Abstract

Quantum reservoir computing (QRC) has emerged as a promising paradigm for harnessing near-term quantum devices to tackle temporal machine learning tasks. Yet, identifying the mechanisms that underlie enhanced performance remains challenging, particularly in many-body open systems where nonlinear interactions and dissipation intertwine in complex ways. Here, we investigate a minimal model of a driven-dissipative quantum reservoir described by two coupled Kerr-nonlinear oscillators, an experimentally realizable platform that features controllable coupling, intrinsic nonlinearity, and tunable photon loss. Using Partial Information Decomposition (PID), we examine how different dynamical regimes encode input drive signals in terms of *redundancy* (information shared by each oscillator) and *synergy* (information accessible only through their joint observation). Our key results show that, near a critical point marking a dynamical bifurcation, the system transitions from predominantly redundant to synergistic encoding. We further demonstrate that synergy amplifies short-term responsiveness, thereby enhancing immediate memory retention, whereas strong dissipation leads to more redundant encoding that supports long-term memory retention. These findings elucidate how the interplay of instability and dissipation shapes information processing in small quantum systems, providing a fine-grained, information-theoretic perspective for analyzing and designing QRC platforms.

## 1. Motivation and Introduction

Reservoir computing (RC) is a computational paradigm that harnesses the intrinsic dynamics of complex systems to process time-dependent inputs efficiently [1,2,3]. Unlike conventional recurrent neural networks (RNNs), RC requires training only at the readout layer, circumventing expensive weight-update procedures on internal nodes [4]. Quantum reservoir computing (QRC) extends these ideas to quantum platforms, leveraging quantum superposition and entanglement to amplify the dimensionality of the feature space and potentially enhance computational capabilities [3,5]. Early demonstrations of QRC have shown promise in tasks like time-series prediction, classification, and memory capacity estimation, and ongoing efforts explore a range of theoretical and experimental strategies for improving performance [6,7,8,9,10,11,12,13,14,15].

Recent QRC research has primarily focused on many-body quantum systems, where quantum phase transitions are suspected to boost computational expressivity [16,17]. While numerical studies reveal intriguing heuristics, such as enhanced memory capacity near critical points, designing optimal quantum reservoirs remains an open question, partly due to the complexity of analyzing large quantum systems. Here, we adopt a complementary approach by studying a *pair of coupled Kerr-nonlinear oscillators*, a minimal yet experimentally realizable quantum platform [18]. This system exhibits rich dynamical behaviors, including dynamical instability (bifurcation) and dissipation due to photon loss that can be precisely tuned. As such, it offers a tractable yet nontrivial testbed for exploring how quantum correlation, instability, and dissipation govern quantum information processing.

To dissect how these coupled oscillators encode incoming signals, we draw on information-theoretic concepts from neuroscience, where measures of synergy and redundancy helped analyze how neural networks collectively encode stimuli [19,20,21,22]. As the traditional mutual information metric fails to separate out redundant and synergistic contributions to information encoding, we employ *Partial Information Decomposition* (PID) [23], which partitions the total information into three components: *redundancy*, capturing information that both oscillators share; *unique information*, provided by each oscillator individually; and *synergy*, arising only when both oscillators are observed together. This perspective provides a fine-grained view of the internal encoding structure of the reservoir.

To connect the system’s dynamics to its information-encoding strategy, we combine numerical simulations with non-equilibrium mean-field theory based on the Keldysh formalism [24], focusing on how small external perturbations propagate through the system. In particular, we study how the coupling strength, frequency detuning, and photon loss rate influence the system’s response, and then show how these distinct dynamical regimes lead to different synergy and redundancy profiles in the oscillators’ outputs.

Our main findings reveal that near a critical coupling strength leading to dynamical bifurcation, the system transitions from predominantly redundant encoding to a regime featuring significant synergistic information. This synergistic behavior arises from the interplay between fast *collective* oscillations and overdamped *soft* modes. We show that increasing dissipation suppresses quantum correlations and promotes highly redundant encoding modes. In contrast, near the onset of dynamical instability, synergy is amplified and enriches short-term responsiveness, improving short-term memory retention. Taken together, these results highlight how dissipation and dynamic instability in a minimal system can steer a quantum reservoir toward redundant or synergistic processing, each regime benefiting different computational tasks.

In relation to recent proposals using two coupled Kerr-nonlinear oscillators as quantum reservoirs [25,26], which highlight the roles of dissipation for fading memory and moderate coupling for richer dynamics, our work differs by focusing on PID to examine how synergy and redundancy influences the reservoir’s memory capacity. This complements prior findings to show that critical points in Kerr dynamics can shift encoding from redundant to synergistic regimes. Meanwhile, single Kerr oscillators with large Hilbert spaces [13,27] highlight how dimensionality alone can serve as a computational resource, but our key question of *whether the whole can exceed the sum of its parts* requires at least two coupled oscillators for emergent synergistic encoding. Lastly, although [28] studies larger arrays of Kerr oscillators and demonstrates the near-bifurcation enhancements of nonlinear memory and employ higher-order cumulant expansions to handle higher photon numbers, we limit ourselves to at most a second-order expansion in a parameter regime where it remains accurate, focusing on the PID-based insights rather than reservoir benchmarks and thus complementing other findings in the literature.

To guide the reader, this paper is organized as follows. Section 2 introduces the coupled Kerr-oscillator model and the relevant information-theoretic measures, including PID and quantum mutual information. Section 3 then presents our core numerical findings on synergy and redundancy, comparing fully quantum dynamics with both mean-field and cumulant expansion analyses. We also discuss the mechanisms driving redundant and synergistic encoding and examine how dissipation influences these encoding modes. In Section 3.4, we connect these insights to the quantum reservoir’s memory capacity. Finally, we conclude in Section 4 by summarizing our results and outlining directions for future research. A pedagogical overview of PID can be found in Appendix A and Appendix B, and the details of Keldysh approach to linear response analysis is provided in Appendix C.

## 2. Quantum Model, Relevant Information Measures, and Performance Metrics

We first introduce our quantum system to study the interplay of instability and dissipation and their influence on modes of information encoding. Then, we outline the key goals of our study, and introduce information measures and metrics to characterize our quantum systems, and finally describe the numerical methods to study them.

### 2.1. Model

We consider a minimal model that can exhibit dynamical transitions from simple to more complex dynamics, and also support both redundant and synergistic modes of information encoding: a pair of coupled Kerr-nonlinear oscillators. Such systems are well studied in cavity quantum electrodynamics and nonlinear optics, where the interplay of nonlinearities and external driving yields rich dynamical behavior [29,30,31,32]. By focusing on just two coupled cavities, each supporting a single mode at a particular resonance frequency, we avoid the complexity of many-body phases that arise in the thermodynamic limit, thus isolating the essential ingredients needed to explore the onset of coordinated information encoding behaviors in quantum systems. The schematic diagram of this setup is presented in Figure 1.

We work in a frame rotating at the driving frequency $\omega_{F}$, and the corresponding Hamiltonian is given by(1)$$\hat{H}\left(t\right)=J\left({\hat{a}}_{1}^{†}{\hat{a}}_{2}+{\hat{a}}_{2}^{†}{\hat{a}}_{1}\right)+\sum_{i=1,2}\left(\Delta_{i}{\hat{a}}_{i}^{†}{\hat{a}}_{i}+\frac{1}{2}U_{i}{\hat{a}}_{i}^{†2}{\hat{a}}_{i}^{2}+F\left(t\right)\left({\hat{a}}_{i}^{†}+{\hat{a}}_{i}\right)\right),$$
where ${\hat{a}}_{i}$ and ${\hat{a}}_{i}^{†}$ are the annihilation and creation operators for the ith cavity mode, respectively. The parameter *J* governs the coupling strength between the two cavities, enabling photon tunneling and collective mode formation [30]. The detuning $\Delta_{i}=\omega_{i}-\omega_{F}$ measures the offset of the ith cavity’s resonance frequency $\omega_{i}$ from the driving frequency. The nonlinear Kerr coefficient $U_{i}$ characterizes the anharmonicity of each mode which introduces photon-photon interactions essential for generating nonclassical states. And F(t) is a common external drive that contains time-dependent information applied to both cavities.

To observe transitions between redundant and synergistic information encoding, the two cavities must differ in their nonlinear properties. In particular, having distinct Kerr coefficients ($U_{1}\neq U_{2}$) breaks symmetry and enables nontrivial interactions between the modes. With these minimal ingredients (a coherent drive, a tunable coupling, and carefully chosen nonlinearities), this system provides a controlled setting to study the fundamental mechanisms underlying both redundant and synergistic coding in interacting quantum systems.

To incorporate noise and dissipation, we consider the time evolution of an open quantum system weakly coupled to a Markovian bath. Specifically, we model the dynamics of the system’s density matrix $\hat{\rho }\left(t\right)$ using the Lindblad master equation [33]:(2)$$\frac{d}{dt}\hat{\rho }\left(t\right)=L\hat{\rho }\left(t\right)=-i\left[\hat{H}\left(t\right),\hat{\rho }\left(t\right)\right]+\sum_{i=1,2}2\gamma_{i}\;D\left[{\hat{a}}_{i}\right]\hat{\rho }\left(t\right),$$
where $\gamma_{i}$ is the photon decay rate of the *i*th cavity associated with the Lindblad superoperator describing single-photon loss, $D[{\hat{a}}_{i}]$, which acts on the density matrix as(3)$$D\left[{\hat{a}}_{i}\right]\hat{\rho }={\hat{a}}_{i}\hat{\rho }{\hat{a}}_{i}^{†}-\frac{1}{2}\left\{{\hat{a}}_{i}^{†}{\hat{a}}_{i},\hat{\rho }\right\}.$$

In our simulations, we take $\gamma_{1}=\gamma_{2}=\gamma$ for simplicity.

For the common time-dependent external driving field F(t), we choose F(t)=s(t)F, where s(t) is a dimensionless, time-dependent signal, and *F* is a characteristic strength of the drive. To ensure that the system’s intrinsic dynamics dominate, we select *F* to be small or comparable to other energy scales, and regard F(t) as a perturbation. In this work, s(t) is taken to be a symmetric telegraph process with s(t)∈{−1,1} [34]. While telegraph noise may not be a directly implementable noise model in all bosonic systems (an open quantum system can couple to telegraph noise if it interacts with a fermionic bath; see [35,36] for related studies), we employ it here as a convenient testing ground. Its well-characterized statistical properties [34] and ease of numerical simulation make it a useful drive model for probing how redundant and synergistic information encoding emerges in quantum dynamics. In Section 3.2, we also compare the results with those obtained using different noise models to assess their generality.

### 2.2. From Quantum to Semiclassical (Mean-Field) Dynamics

Directly simulating the Lindblad equation is practical only when the average photon number is small, as the Hilbert space dimension grows rapidly with photon occupation. In this low-photon regime, we simulate the full quantum dynamics in Equation (Equation 2) to capture all quantum correlations, compute PID and QMI, and analyze how nonclassical effects influence information encoding.

As we increase the driving strength or adjust parameters to reach higher photon-number regimes, the full quantum simulation becomes computationally expensive. In this regime, quantum fluctuations often play a less significant role, and a semiclassical approximation becomes suitable. By factorizing expectation values as $\langle {\hat{a}}_{i}{\hat{a}}_{j}\rangle \approx \langle {\hat{a}}_{i}\rangle \langle {\hat{a}}_{j}\rangle$, the dynamics reduce to coupled nonlinear ordinary differential equations (ODEs) for $\alpha_{i}\left(t\right)=\langle {\hat{a}}_{i}\rangle$:(4)$$\begin{array}{c} \begin{array}{cc} \frac{d}{dt}\alpha_{1} & =-\left(\gamma +i\Delta \right)\alpha_{1}-iJ\alpha_{2}-iU_{1}\alpha_{1}{\left|\alpha_{1}\right|}^{2}-iF\left(t\right),\\ \frac{d}{dt}\alpha_{2} & =-\left(\gamma +i\Delta \right)\alpha_{2}-iJ\alpha_{1}-iU_{2}\alpha_{2}{\left|\alpha_{2}\right|}^{2}-iF\left(t\right). \end{array} \end{array}$$

These ODEs are much easier to solve, allowing us to examine information encoding under conditions where the photon number is large and quantum correlations are negligible.

To bridge the gap between the fully quantum and semiclassical treatments, we also employ a second-order cumulant expansion (see Appendix D). This approach partially restores some quantum correlations while remaining more tractable than the full density-matrix simulation. We expect that in parameter regimes where quantum correlations matter, the cumulant expansion will improve upon the semiclassical approximation but still remain simpler than the full quantum approach.

### 2.3. Characterizing Information Processing

To characterize how our system of coupled Kerr-nonlinear oscillators processes and encodes the *input* signal s(t) into the *output* readouts taken to be(5)Xi(t)≡Re〈a^i(t)〉,
we analyze three complementary figures of merit. First, we use the Partial Information Decomposition (PID) to separate the total information that the output observables encode about the input into *redundant* and *synergistic* components. Second, we employ the quantum mutual information (QMI) to quantify the role of quantum correlations in shaping these encoding modes. Finally, we consider the memory capacity in a quantum reservoir computing (QRC) context to assess how information is retained over time. While PID and QMI directly characterize the system’s response to external inputs without any training procedure, the memory capacity inherently involves a training step to quantify how well past inputs can be reconstructed from the system’s outputs. In this way, all three measures together provide a comprehensive view of the system’s information processing capabilities.

#### 2.3.1. Partial Information Decomposition (PID)

Let *s*, $X_{1}$, and $X_{2}$ be random variables representing the input signal and the measured observables from the two oscillators, respectively. The mutual information $I(s:X_{1},X_{2})$ can be decomposed into redundant, synergistic, and unique components as(6)$$I\left(s:X_{1},X_{2}\right)\;=\;\mathrm{Rdn}+\mathrm{Syn}\;+\;\mathrm{Unq}\left(X_{1}\right)+\mathrm{Unq}\left(X_{2}\right),$$
where Rdn is the redundant information present in both $X_{1}$ and $X_{2}$, $\mathrm{Unq}(X_{i})$ is the unique information contributed solely by $X_{i}$, and Syn is the synergistic information accessible only through the joint knowledge of $X_{1}$ and $X_{2}$.

By constructing the empirical joint distribution $P(s,X_{1},X_{2})$ from simulation data and applying the BROJA-2PID algorithm [37], we isolate Rdn and Syn. This allows us to determine whether the oscillators encode input information redundantly or synergistically, thereby shedding light on their cooperative information processing strategies across different dynamical phases of the system. More detailed discussions and example calculations of PID can be found in Appendix A and Appendix B.

#### 2.3.2. Quantum Mutual Information (QMI)

Although our primary Partial Information Decomposition (PID) analysis focuses on classical correlations between input-output variables, the underlying reservoir dynamics remain fundamentally quantum. To probe the quantum correlations inherent in our system, we compute the *quantum mutual information* (QMI) between the two oscillators.

Let $\rho_{12}$ be the density matrix describing the joint state of the two coupled Kerr oscillators. We partition the system into subsystems 1 and 2, each corresponding to one oscillator. The QMI is then given by(7)$$I\left(1:2\right)\;=\;S\left(\rho_{1}\right)\;+\;S\left(\rho_{2}\right)\;-\;S\left(\rho_{12}\right),$$
where $\rho_{1}={\mathrm{Tr}}_{2}\left(\rho_{12}\right)$ and $\rho_{2}={\mathrm{Tr}}_{1}\left(\rho_{12}\right)$ are the reduced density matrices of each oscillator, and $S\left(\rho \right)=-\mathrm{Tr}\;\left(\rho \ln\rho \right)$ is the von Neumann entropy.

Because each oscillator’s Hilbert space is, in principle, infinite dimensional, we employ a finite photon-number basis up to a cutoff $N_{\mathrm{cutoff}}$ to ensure computational tractability. Namely,(8)$$\rho_{12}\;\approx \;\sum_{n_{1}=0}^{N_{\mathrm{cutoff}}}\sum_{n_{2}=0}^{N_{\mathrm{cutoff}}}c_{n_{1},n_{2}}\;|n_{1}\rangle \langle n_{1}\left|\;⊗\;\right|n_{2}\rangle \langle n_{2}|,$$
where $|n_{i}\rangle$ is the Fock state with $n_{i}$ photons in oscillator *i*, and $c_{n_{1},n_{2}}$ are the matrix elements obtained by time-averaging the steady-state solution of the master equation. We verify that increasing $N_{\mathrm{cutoff}}$ beyond ∼10 does not significantly alter the results within the quantum regime studied in this work, suggesting that the computed QMI converges to a within numerical precision.

To evaluate Equation (Equation 7), we first diagonalize the full two-oscillator density matrix $\rho_{12}$(9)$$\rho_{12}\;=\;\sum_{k}\lambda_{k}\;|\psi_{k}\rangle \langle \psi_{k}|,\;S\left(\rho_{12}\right)\;=\;-\sum_{k}\lambda_{k}\;\ln\lambda_{k},$$
where $\left\{\lambda_{k}\right\}$ and $\left\{|\psi_{k}\rangle \right\}$ are the eigenvalues and eigenbases of $\rho_{12}$, respectively. We then obtain the reduced density matrices for each oscillator by tracing out the other as $\rho_{1}={\mathrm{Tr}}_{2}\left(\rho_{12}\right),$ and $\rho_{2}={\mathrm{Tr}}_{1}\left(\rho_{12}\right).$ Both $\rho_{1}$ and $\rho_{2}$ are similarly diagonalized to compute their von Neumann entropies, $S(\rho_{1})$ and $S(\rho_{2})$. Substituting these into Equation (Equation 7) gives the QMI. The resulting I(1:2) quantifies the *total* correlations between the two oscillators, including both classical and quantum components, and thus complements a classical PID-based analysis.

#### 2.3.3. Memory Capacity (MC)

In addition to instantaneous encoding, we are interested in how the system stores information over time, as is relevant in quantum reservoir computing (QRC). The memory capacity quantifies how well the current state of the reservoir (the outputs $X_{1}\left(t\right)$, $X_{2}\left(t\right)$) retains information about past inputs s(t−τ). By analyzing how the reconstruction error of past inputs varies with the delay τ, we derive a memory measure that complements the PID and QMI analyses.

A high memory capacity suggests that the reservoir not only encodes the input at a given instant but also preserves information over extended periods. Comparing the memory capacity with Rdn and Syn reveals how different dynamical regimes influence both the instantaneous and temporal dimensions of information processing in this system. Details of the MC calculation are provided in Section 3.4.

## 3. Results and Discussion

We now present numerical evidence that coupled Kerr oscillators can encode input signals in either a redundant or synergistic fashion, depending on *J*, Δ, and γ.

### 3.1. Emergence of Synergistic Encoding

We begin by examining how the joint mutual information (MI), $I\left(s:(X_{1},X_{2})\right)$, compares to the individual MIs $I(s:X_{1})$ and $I(s:X_{2})$ when probing our driven-dissipative system. We focus on two representative parameter sets: a *mean-field* regime with larger drive and smaller Kerr nonlinearities, and a *quantum* regime with smaller drive and larger Kerr nonlinearities. In both cases, we fix the detuning and damping at Δ=−2 and γ=0.5. Concretely, in the mean-field case, we take F=2.0, $U_{1}=6.25\times 10^{-3}$, and $U_{2}=2U_{1}$, while in the quantum regime, we take F=0.2, $U_{1}=4.0$, and $U_{2}=2U_{1}$.

#### 3.1.1. Synergy from Total Mutual Information Consideration

Figure 2 compares $I\left(s:(X_{1},X_{2})\right)$ with $I(s:X_{1})$ and $I(s:X_{2})$, revealing that$$I\left(s:(X_{1},X_{2})\right)\;>\;I\left(s:X_{1}\right)\;+\;I\left(s:X_{2}\right),$$
in the mean-field dynamics regime. This information excess indicates that measuring both $X_{1}$ and $X_{2}$ jointly can reveal strictly more information about the external drive signal s(t) than either observable alone, suggesting a potential *synergistic* encoding mechanism.

#### 3.1.2. Transition from Redundant to Synergistic Encoding

To further investigate whether this information surplus really originates from synergistic effects (rather than unique information in each oscillator), we perform Partial Information Decomposition (PID) according to Equation (Equation 6). For clarity, we normalize synergy and redundancy by $I\left(s:(X_{1},X_{2})\right)$, respectively,$${\mathrm{Syn}}_{\mathrm{norm}}=\frac{\mathrm{Syn}}{I\left(s:(X_{1},X_{2})\right)},\;{\mathrm{Rdn}}_{\mathrm{norm}}=\frac{\mathrm{Rdn}}{I\left(s:(X_{1},X_{2})\right)}.$$

In Figure 3, we compare the normalized synergy (left) and redundancy (right) across three regimes (mean-field, second-order cumulant, and fully quantum) at fixed detuning and dissipation. As we sweep the coupling strength *J* from small to large, a pronounced synergy peak emerges near J≈|Δ|=2, marking a transition from predominantly redundant encoding to notably higher synergy (near J≃|Δ|). In the quantum regime, stronger quantum correlations bias the encoding scheme slightly toward redundancy even near the peak. In contrast, the second-order cumulant approach captures the partial quantum correlations that lie between the mean-field regime (where correlations are suppressed) and the fully quantum regime (where all orders of correlations may appear). This second-order cumulant dynamics provides an approximate interpolation for regimes where quantum correlations are significant but not too strong (we set $\Delta =-2,F=0.5,\gamma =0.5,U_{1}=0.2,U_{2}=2U_{1}$ to represent a dynamical regime with non-negligible quantum correlations, motivating the use of second-order cumulants); see Appendix D.

### 3.2. Underlying Mechanisms Enhancing Synergistic Coding: The Role of Soft and Fast Modes

In this section, we explain the sharp increase in synergy observed near J≃|Δ|, and attribute this behavior to the interplay between soft and fast modes in the coupled Kerr oscillators. Specifically, we demonstrate that the dominance of fast modes, enabled by the overdamping of soft modes, enhances coherent collective dynamics, leading to an increase in synergistic information.

#### 3.2.1. Soft Modes and Potential Landscape Flatness

In the mean-field approximation without external drive (F=0), the coupled Kerr oscillators evolve in the following effective potential (see Appendix C), which captures the interplay of detuning, Kerr nonlinearities, and coupling(10)$$V\left({\vec{\alpha }}_{c},{\vec{\alpha }}_{c}^{*}\right)\;=\;\sum_{j=1,2}\left(\Delta \;{\left|\alpha_{j,c}\right|}^{2}+\frac{1}{4}U_{j}\;{\left|\alpha_{j,c}\right|}^{4}\right)\;+\;J\left(\alpha_{1,c}\;\alpha_{2,c}^{*}+\alpha_{2,c}\;\alpha_{1,c}^{*}\right),$$
where $\Delta_{1}=\Delta_{2}=\Delta$ and ${\vec{\alpha }}_{c}=\left[\alpha_{1,c},\alpha_{2,c}\right]$. At J=|Δ|, the Hessian of *V* evaluated at the steady-state solution $\alpha_{1,c}=\alpha_{2,c}=0$ develops zero eigenvalues, corresponding to flat or marginal directions. These flat directions represent soft modes, characterized by near-zero oscillation frequencies. Specifically, when evaluated at the steady state $\alpha_{1,c}=\alpha_{2,c}=0$, the Hessian matrix of this effective potential (Equation 10) calculated in terms of the vector (Re(α1,c),Im(α1,c),Re(α2,c),Im(α2,c)) is(11)$$H\left[V\left(\vec{\alpha },{\vec{\alpha }}^{*}\right)\right]{|}_{\vec{\alpha },{\vec{\alpha }}^{*}=0}=2\left(\begin{array}{cccc} \Delta & 0 & J & 0\\ 0 & \Delta & 0 & J\\ J & 0 & \Delta & 0\\ 0 & J & 0 & \Delta \end{array}\right).$$

The eigenvalues of this Hessian are{2(Δ+J),2(Δ+J),2(Δ−J),2(Δ−J)}.

For the relevant parameter regime in which Δ<0 and J>0, exactly when J=|Δ| that two flat directions (zero eigenvalues) emerge, and the other two directions are unstable at the second order (negative eigenvalues); see Figure 4.

#### 3.2.2. Overdamping of Soft Modes at J=|Δ|

Including dissipation with a rate γ can transform the nearly flat directions into overdamped dynamics as follows. Following the Keldysh formalism [38,39,40] (see Appendix C), the retarded Green’s function shows poles of the form(12)$$\omega_{s}\;=\;\pm \;\left|J-|\Delta |\right|\;-\;i\;\gamma ,\;\omega_{f}\;=\;\pm \;\left|J+|\Delta |\right|\;-\;i\;\gamma ,$$
where $\omega_{s}$ (slow) and $\omega_{f}$ (fast) label the respective branches. Exactly at J=|Δ|, the real part of $\omega_{s}$ *vanishes*, leaving only −iγ, indicating an overdamped relaxation to the steady state. In contrast, the fast modes remain oscillatory with frequencies Re(ωf)=|J+|Δ||. Consequently, at J=|Δ|, the dynamics are dominated by the coherent oscillations of the fast modes, as the soft modes contribute only non-oscillatory relaxation.

#### 3.2.3. Coherence-Driven Synergy Enhancement

When the dissipation rate is comparable to the oscillatory frequency of the fast modes (γ∼Re(ωf)), the dominance of fast modes at J=|Δ| reduces competition between oscillatory frequencies (soft modes disappear and only fast modes persist) and enhances the coherence of the system’s collective response. More specifically, following the discussion in Appendix C, one can consider the relaxation dynamics of the small perturbation $\delta \alpha_{c}\left(t\right)$ around the steady state. It follows that the relaxation dynamics of each observable at site *j* can be expressed as(13)Reδαj,c(t)=cj,se−γtcos(Re(ωs)t)+cj,fe−γtcos(Re(ωf)t),
where $c_{j,s}$ and $c_{j,f}$ are initial-perturbation-dependent mode amplitudes. When Re(ωs)→0, the soft mode contribution simplifies to pure exponential decay, and the dissipative dynamics are dominated by the single oscillatory frequency $\omega_{f}$ of the fast modes.

This coherence relaxation eliminates competing oscillations and minimizes overlapping contributions between modes, reducing redundancy. Also, the collective dynamics driven by the fast modes encode information that cannot be captured by any single oscillator alone, thereby enhancing synergistic information. This explains the peak in synergy observed near J≃|Δ|.

Figure 5 illustrates how the slow-mode poles (orange) move to the imaginary axis at J=|Δ|, marking the *disappearance of competing oscillatory timescales*. In this near-critical regime, the relaxation dynamics become dominated by the underdamped (oscillatory) contributions of the fast modes, resulting in coherent dissipation. It is important to note that this result pertains to the regime where γ∼|Re(ωf)|≫|Re(Ωs)|.

#### 3.2.4. Generality of the Transition to Synergistic Behavior

The observed synergy peak at J≃|Δ| is not specific to the type of input signal driving the system. To demonstrate this, we perform numerical simulations of the master equation describing quantum dynamics with the input signal s(t) sampled from a uniform distribution in the interval [−1,1], and uncorrelated in time. The results, shown in Figure 6, confirm that the transition in redundant/synergistic behavior persists at J=|Δ|, regardless of the statistical properties of the input.

These results emphasize that the transition to synergistic encoding at J≃|Δ| is an intrinsic feature of the system’s response dynamics, driven by the dominance of underdamped fast modes, and not much by the input signal properties. In the following section, we explore how increasing γ impacts the system’s encoding behavior, showing that fast relaxation towards the steady state progressively shifts the system from synergistic to redundant encoding.

### 3.3. Large Dissipation Leads to Redundant Encoding

As the damping rate γ increases, the dynamics progressively shift toward overdamped relaxation for both slow and fast modes. This transition leads to a steady-state regime where the two cavities become nearly identical, resulting in redundant coding of the input information. While the emergence of soft modes and the dominance of fast modes at J≃|Δ| enhance synergy, increasing γ gradually suppresses this effect, shifting the system toward a more redundant encoding regime dominated by rapid overdamped relaxation towards the steady state.

This behavior is particularly evident in the quantum regime, where the system approaches a product state at large γ, rendering the two oscillators effectively independent. As shown in Figure 7, increasing γ from the baseline value of γ=0.5 (as seen in Figure 2 and Figure 3) results in a clear reduction in quantum correlations. Panel (a) of Figure 7 quantifies this through the time-averaged quantum mutual information (QMI), which decreases as γ increases. Notably, peaks in QMI at J≃|Δ| coincide with peaks in classical mutual information (MI) as shown in panel (b). This observation aligns with the intuition that higher quantum correlations often translate to enhanced classical information encoding near the critical coupling.

Figure 7c,d further illustrate this transition by showing how absolute synergy diminishes and absolute redundancy grows with increasing γ. At low γ, the dynamics favor (weakly) synergistic encoding. At high γ, however, the system becomes dominated by (highly) redundant encoding, with both cavities responding similarly and independently of each other.

### 3.4. Memory Capacity of Synergistic and Redundant Encoding

We close our discussion by analyzing the performance of the coupled Kerr oscillators as a quantum reservoir, focusing on their capacity to retain and process temporal information. This memory capacity benchmark highlights how the synergistic and redundant behaviors identified earlier influence practical tasks such as time-series memorization.

#### 3.4.1. Short-Term Memory Task

To quantify memory capacity, we train the system to recall past input signals using a short-term memory task. The input signal s(t) is sampled from a uniform distribution in the interval [−1,1] and is uncorrelated in time, and the target time series ${\bar{y}}_{n}\left(t\right)$ corresponds to the input signal at a previous time step:(14)$${\bar{y}}_{n}\left(t\right)=s\left(t-n\Delta t\right),$$
where Δt=0.01 is the time step. The output observables Xi(t)=Re(〈a^i(t)〉) and Yi(t)=Im(〈a^i(t)〉) are used as feature vectors (here, we use more output readouts than in the previous sections since this input signal is more difficult to fit with fewer feature vectors). We construct an output vector $\vec{X}\left(t\right)$: (15)$$\vec{X}\left(t\right)={\left(X_{1}\left(t\right),X_{2}\left(t\right),Y_{1}\left(t\right),Y_{2}\left(t\right),1\right)}^{⊤},$$
and fit the target ${\bar{y}}_{n}\left(t\right)$ using a weight matrix W via standard linear regression with Tikhonov regularization:(16)$$\hat{y}\left(t\right)=W^{*}\vec{X}\left(t\right),$$
where the optimal weights $W^{*}$ minimize the mean square error (MSE) during training:(17)$$\mathrm{MSE}\left(W,\lambda \right)=\left|\right|\bar{y}-\hat{y}{\left|\right|}^{2}+{\lambda ||W||}^{2}.$$

For simplicity, we set λ=0. After training, the memory capacity [1] for delay step *n* is evaluated as(18)$$\mathrm{MC}\left(n\right)=\frac{{\mathrm{cov}}^{2}\left({\bar{y}}_{n},{\hat{y}}_{\mathrm{pred}}\right)}{\sigma^{2}\left({\bar{y}}_{n}\right)\sigma^{2}\left({\hat{y}}_{\mathrm{pred}}\right)},$$
where 0≤MC(n)≤1, with MC(n)=1 indicating perfect recall.

#### 3.4.2. Tuning *J* Towards the Critical Coupling Strength

Figure 8 shows the memory capacity as *J* approaches the critical coupling J=|Δ|=2. At smaller values of *J*, the memory capacity remains low across the delay steps. However, as *J* approaches |Δ|, the system exhibits a notable change in behavior: the memory capacity for short delays (small *n*) increases significantly, while the memory capacity for longer delays (large *n*) decays more rapidly. This behavior reflects the trade-off between short-term and long-term memory as the system transitions to the synergistic regime dominated by fast modes.

We note that, for low dissipation (e.g., γ=0.5), the system may appear to retain information about initial conditions at long delays *n*, indicating that the fading memory property might not be fully realized. While a more comprehensive benchmark such as the information processing capacity (IPC) [41] could more definitively confirm or refute fading memory in this low-dissipation regime, our current goal is to investigate synergy and redundancy in a small quantum reservoir rather than to perform an exhaustive reservoir-computing benchmark. Instead, we emphasize qualitative trends in short- vs. long-term memory. A more detailed, IPC-based study would be an interesting direction for future work to fully study the computational potential of near-critical quantum reservoirs in this system.

#### 3.4.3. Impact of Dissipation γ

Figure 9 illustrates how dissipation (γ) affects memory capacity at the critical coupling J=|Δ|=2. As γ increases, the memory capacity can be attributed to the enhanced stability (more rapid relaxation towards) in the reservoir’s steady-state dynamics. Dissipation suppresses oscillatory behavior and stabilizes the reservoir’s response. This stabilization corresponds to a regime of highly redundant encoding, where information is stored near the steady state across somewhat identical subsystems.

Interestingly, as shown in Figure 8, while higher γ leads to improved total memory capacity, the decay rate of memory capacity in this redundant regime, MC(n)∼exp(−Γn), remains approximately constant across different dissipation rates. This suggests that dissipation uniformly governs the loss of correlations over time. Differences in memory capacity at high γ primarily arise from the proportionality factor in Equation (Equation 18). As dissipation increases, the variance in the reservoir’s output prediction decreases, reflecting rapid stabilization to the steady state. This reduced variance amplifies the initial memory capacity but does not alter the exponential decay rate of correlations.

These results highlight a subtle interplay between dissipation, memory capacity, and encoding modes. Near criticality at J=|Δ|, the system exhibits synergistic behavior, where collective dynamical response dominates, enabling the reservoir to respond sensitively to recent input signals. This synergistic encoding boosts short-term memory retention but comes at the expense of long-term storage, as the system’s ability to retain correlations with far past inputs diminishes rapidly due to sensitivity to perturbation.

At higher dissipation rates γ, the system tends to encode inputs more redundantly and relaxes more quickly to a steady state. Interestingly, as illustrated in Figure 9, both short- and long-term memory capacities, at criticality, increase with γ for a simple uniformly random input memorization task, even though one might initially expect a trade-off. The underlying reason could be that the system has not yet fully achieved the fading memory regime, so further investigations, potentially through more comprehensive benchmarks such as IPC analysis, are needed to confirm how dissipation precisely balances short-term responsiveness and long-term stability right at a critical point.

Also from the perspective of quantum reservoir design, dissipation plays a critical role in ensuring the reservoir exhibits the fading memory property, where the influence of past inputs gradually diminishes, which is a necessary condition for designing operational reservoir computing [9,10].

## 4. Conclusions and Outlook

In this work, we explored coupled Kerr-nonlinear oscillators as a model open quantum system for studying modes of information processing in a small quantum reservoir computing (QRC) platform. By employing Partial Information Decomposition (PID) and analyzing memory retention tasks, we investigated how the interplay of dynamical instability and dissipation governs the encoding of input information into redundant or synergistic modes. These encoding schemes play a crucial role in determining the reservoir’s performance in processing and retaining temporal data.

Our findings reveal several key insights. First, near the critical coupling strength J=|Δ|, the system transitions from predominantly redundant to synergistic encoding. This transition is driven by the dynamics of coherent oscillation modes that dominate as slow modes (soft modes) become overdamped. These collective dynamics enable the reservoir to process information synergistically, boosting short-term memory retention. Importantly, this synergy is robust across different input signals, including telegraph processes and uniform uncorrelated noise, suggesting that the observed transition is an intrinsic feature of the system’s bifurcation near J=|Δ|. This dynamic instability was elucidated through the non-equilibrium (Keldysh) field-theoretic analysis, which highlighted how the disappearance of soft modes amplifies the dominance of fast, coherent modes.

Dissipation (γ) plays an important role in shaping information encoding and memory capacity for our reservoir near criticality. At low γ, synergistic encoding enables collective processing and enhances short-term memory retention, as the reservoir’s dynamics are sensitive to recent input signals. However, as γ increases, dissipation suppresses coherent dynamics, rapidly driving the system toward redundant encoding. In this regime, large dissipation stabilizes encoding by enforcing redundant representations near the steady state, enabling the reservoir to retain information about far-past inputs at the expense of sensitivity to recent input information.

The connection between encoding modes and memory capacity reveals a trade-off: synergistic encoding favors short-term memory retention but is less suited for long-term storage due to its sensitivity to perturbations. Redundant encoding, on the other hand, sacrifices responsiveness to recent inputs but improves long-term retention. From a quantum reservoir design perspective, dissipation ensures the reservoir exhibits the fading memory property necessary for effective reservoir design [9,42]. By carefully tuning dissipation, one can hope to optimize the balance between short-term responsiveness and long-term stability and tailor the reservoir to specific computational tasks, aligning with the recent work on engineered dissipation as computational resources in quantum systems [10,43,44].

Coupled Kerr-nonlinear oscillators provide a minimal yet insightful testbed for analyzing transitions between encoding modes driven by dissipation and dynamical instability. Extending this framework to other quantum systems with dynamical phase transitions, such as Bose–Hubbard models [40], driven-dissipative platforms [45], and spin systems [13,16], could deepen our understanding of how dissipation and instability shape encoding dynamics in systems with more complex phase spaces. Another interesting direction is to develop a rigorous definition of quantum synergy. Although this work uses Partial Information Decomposition (PID) to analyze classical information derived from quantum observables, incorporating quantum correlations could provide a more comprehensive view of how coherence and other quantum effects either enhance or constrain information encoding, in line with [7,8,13]. Such development would refine our understanding of synergy and redundancy in small quantum systems from a more quantum information theoretic viewpoint.

In practice, realistic quantum devices inevitably face various noise sources and control imperfections. While these perturbations may shift the crossover point between redundant and synergistic encoding or relocate the system’s critical point, our results show that this crossover in the modes of encoding persists over a wide parameter range. With advanced experimental techniques, such as dynamical decoupling and error mitigation in quantum optics, we expect that the essential physics behind this crossover can still be realized in real experimental systems.

Lastly, it is important to recognize that the results presented here, like much of the prior work, assume perfect readout without the presence of shot noise. As system sizes scale, however, readout processes may suffer from exponential concentration phenomena, requiring exponentially many measurement shots to estimate input-dependent readouts accurately as discussed in [11]. This limitation presents a significant barrier to scalability. In larger quantum reservoir systems, future studies must address how synergy and redundancy behave under realistic measurement constraints. Incorporating the effects of finite measurement precision into the framework of encoding dynamics could lead to more scalable and experimentally feasible designs for quantum reservoirs.

## Figures and Tables

**Figure 1 entropy-27-00088-f001:**
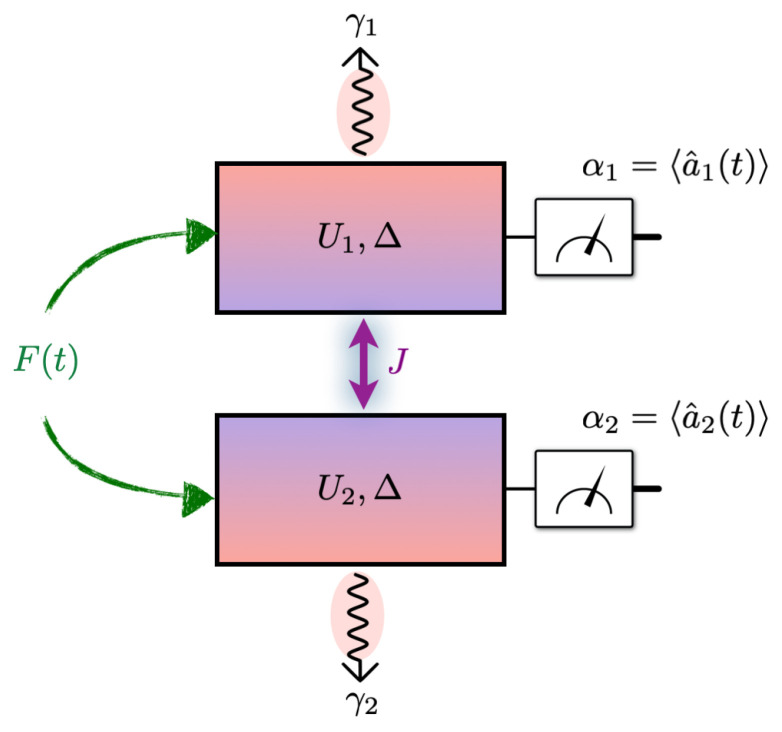
Schematic of two coupled Kerr-nonlinear oscillators. Each cavity *i* features a Kerr nonlinearity $U_{i}$ and a photon loss rate $\gamma_{i}$. A time-dependent drive F(t) (green arrows) injects identical signals into both cavities, while the coherent tunneling of strength *J* (violet arrow) couples the two modes. We measure the mean fields $\alpha_{i}=\langle {\hat{a}}_{i}\left(t\right)\rangle$ to probe the system’s response. The Hamiltonian is specified by Equation (Equation 1), and the Lindblad Equation (Equation 2) governs this driven-dissipative dynamics. We assume both cavities have the same detuning Δ from the drive frequency. This work investigates how the readouts $\alpha_{i}\left(t\right)$ encode the time-dependent drive F(t) across different dynamical regimes of the coupled Kerr oscillators.

**Figure 2 entropy-27-00088-f002:**
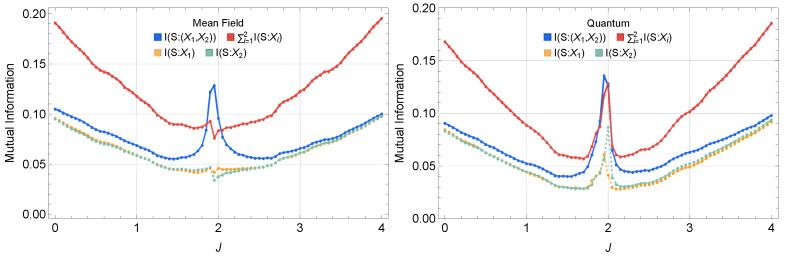
Classical mutual information $I\left(s:(X_{1},X_{2})\right)$, compared to $I(s:X_{1})$ and $I(s:X_{2})$ in (**Left**) a mean-field regime and (**Right**) a quantum regime. In the mean-field regime, $I\left(s:(X_{1},X_{2})\right)$ exceeds $I(s:X_{1})$ or $I(s:X_{2})$ alone near J=|Δ|, hinting at synergy. On the other hand, in the quantum regime, $I\left(s:(X_{1},X_{2})\right)$ is comparable to, but not always exceeding, the sum of $I(s:X_{1})$ and $I(s:X_{2})$.

**Figure 3 entropy-27-00088-f003:**
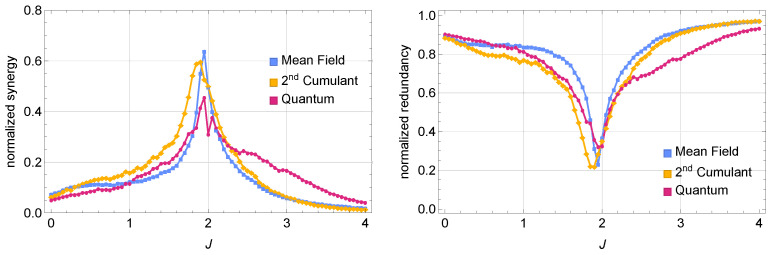
(**Left**) Normalized synergy vs. the coupling *J*. (**Right**) Normalized redundancy vs. the coupling *J*. A pronounced peak near J≈|Δ| marks the crossover from predominantly redundant to more synergistic encoding. In the fully quantum description, enhanced quantum correlations can favor redundancy even at the transition, whereas second-order cumulants interpolate between mean-field and quantum descriptions.

**Figure 4 entropy-27-00088-f004:**
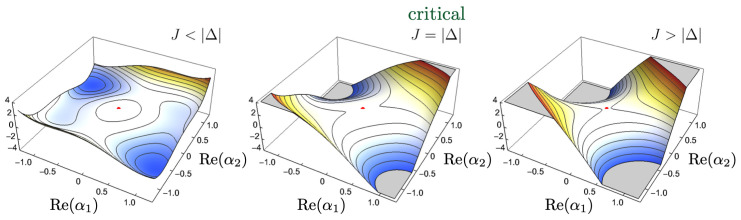
Effective potential around the steady state $\alpha_{1,c}=\alpha_{2,c}=0$ (red dot) in the Δ<0,J>0 regime, projected onto Im(α1)=Im(α2)=0. (**Left**) When J<|Δ|, the steady state is weakly unstable in all directions, with no soft modes present, and the system predominantly encodes inputs redundantly. (**Center**) At the critical point J=|Δ|, flat directions appear, marking the onset of soft modes. In this near-critical regime, collective oscillations enhance synergistic encoding. (**Right**) For J>|Δ|, the potential deforms into a saddle, with two stable and two unstable directions. Here, the soft modes again disappear, and the system encodes inputs redundantly.

**Figure 5 entropy-27-00088-f005:**
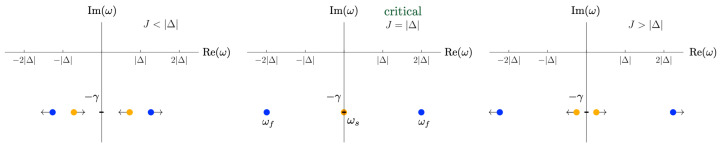
Retarded Green’s function poles (A16) in the complex-frequency plane as *J* increases, illustrating the evolution of slow modes $\omega_{s}$ (orange dots) and fast modes $\omega_{f}$ (blue dots). For J<|Δ|, both slow and fast modes coexist, and the system tends to encode inputs more redundantly. Near the critical point J=|Δ|, the real part of $\omega_{s}$ approaches zero, indicating a disappearance of slow collective oscillations and an overdamped decay. In this near-critical regime, collective oscillations due to underdamped fast modes dominate and enhance synergistic encoding. For J>|Δ|, the slow modes shift away from zero frequency, reducing synergy and transitioning the system back toward redundant encoding. This interplay between slow and fast modes governs how the system transitions from predominantly redundant to synergistic processing and back again as *J* crosses the critical point.

**Figure 6 entropy-27-00088-f006:**
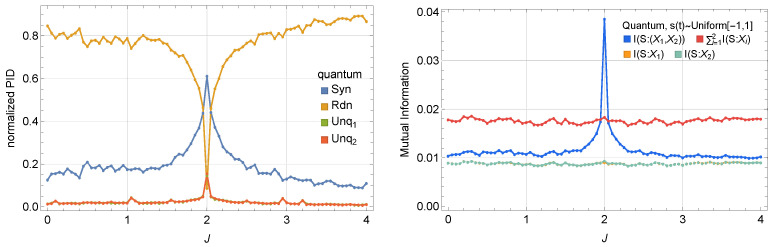
Impact of uniform, uncorrelated noise input on information encoding at the quantum regime (Δ=−2, γ=0.5, and F=0.2). (**Left**) Normalized PID with uniform noise input. (**Right**) Comparison of total MI and partial MI contributions. The left panel shows a clear transition to synergistic encoding near J=|Δ|. The right panel compares $I\left(S:(X_{1},X_{2})\right)$ and $I\left(S:X_{1}\right)+I\left(S:X_{2}\right)$, highlighting the emergence of synergy near the critical coupling.

**Figure 7 entropy-27-00088-f007:**
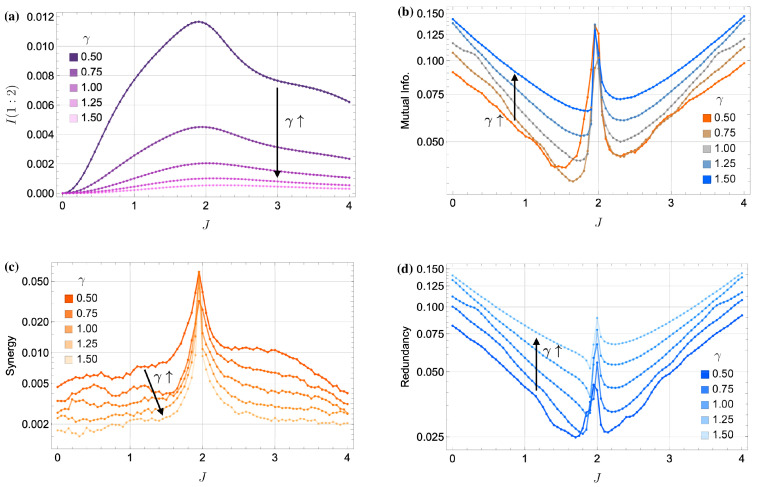
Impact of increasing γ on information metrics in the quantum regime (Δ=−2, F=0.2). (**a**) Time-averaged quantum mutual information (QMI) between the two oscillators as a function of *J* for different γ. (**b**) Classical mutual information (MI) between the input signal and the outputs vs. *J*. (**c**) Absolute synergy vs. *J*. (**d**) Absolute redundancy vs. *J*. These plots illustrate the transition from low-synergistic to high-redundant encoding with increasing γ. At large dissipation, the two oscillators approach a product state, indicated by low QMI. Interestingly, despite the redundancy dominating at higher γ, the total mutual information near J=|Δ| remains approximately constant.

**Figure 8 entropy-27-00088-f008:**
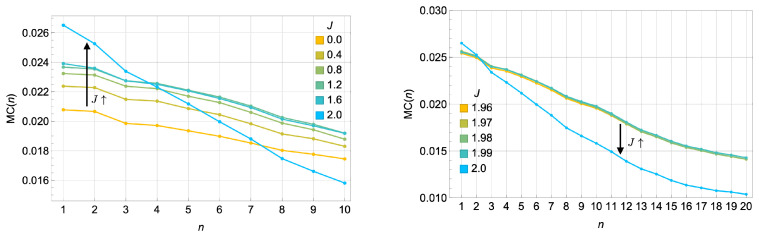
(**Left**) Memory capacity MC(n) for *n* = 1–10 and J∈[0,2], showing an increase in short-term memory capacity as J→|Δ|. (**Right**) Memory capacity for *n* = 1–20 and J∈[1.96,2], showing that the long-term capacity drops as J→|Δ|. Parameters: Δ=−2, γ=0.5, F=0.2. These results are averaged over 50 input realizations.

**Figure 9 entropy-27-00088-f009:**
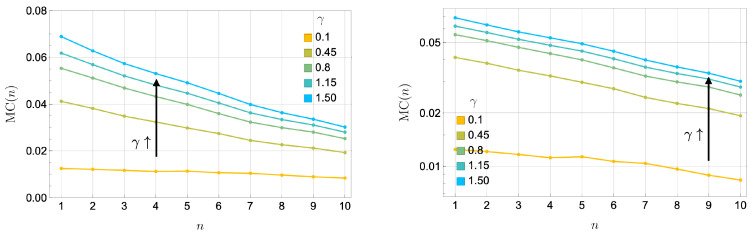
Memory capacity at the critical coupling J=|Δ|=2 as γ increases. (**Left**) Linear scale plot: MC(n) vs. *n*. (**Right**) Log scale plot of MC(n)∼exp(−Γn). The left panel shows the average memory capacity (100 realizations). The right panel illustrates an exponential decay in MC(n). The exponential decay rate of the two-time correlation in the memory capacity, Γ, remains approximately constant as γ increases, indicating that dissipation uniformly governs the loss of correlations across different dissipation rates. Differences in memory capacity primarily arise from the proportionality factor, with larger γ leading to a smaller variance in the output prediction in the denominator of Equation (Equation 18), as the output rapidly stabilizes to the steady state. This rapid stabilization corresponds to highly redundant encoding, and in turn enhances the total memory capacity for memorizing uniformly random input time series. Notably, in this redundant coding regime, the highly dissipative dynamics improve the quantum reservoir’s memory capacity.

## Data Availability

The codes and datasets used in this work are available from the corresponding author upon reasonable request.

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
