# Peer review of "Dissipation Alters Modes of Information Encoding in Small Quantum Reservoirs near Criticality"

_entropy, 2025, doi:10.3390/e27010088_

Round 1
Reviewer 1 Report
Comments and Suggestions for Authors
The authors propose to use tools from information theory to analyze the input response of a quantum reservoir computer. In particular, they study a reservoir made of a pair of coupled Kerr-nonlinear oscillators with coherent drive and single-photon losses. The idea of this proposal is original and the presentation is good, but there are a few aspects that need to be addressed before accepting this paper for publication.
1) The conclusions of Section 3.4 are based on the small values of the MC shown in Figs. 8 and 9. Although trends can be observed, I think it is difficult to draw firm conclusions since the noise level of the MC for long n seems to be around 10^{-2}. That is, the spurious contributions of the MC for very large n seem to be of the same order of magnitude as the contributions for small n. My concern is that the system seems to have a very weak fading memory for low values of \gamma such as 0.5 (where most of the results are discussed), retaining information about the initial conditions and not being a good reservoir for any task. Intuitively, the fading memory must be there, since dissipation works, but the effect could be so small that the system is useless for information processing.
I suggest that a stronger demonstration of the input processing capabilities of the system be included for low \gamma. One possibility could be the calculation of the information processing capacity (Dambre et al. Sci. Rep. 2012). This benchmark would saturate up to the total number of output nodes if the system has fading memory (and linear independent nodes), regardless of this number of nodes.
2) Besides, I think that the conclusions between lines 360-366 should be modified. In these paragraphs, a trade-off between short and long-term memory is assumed when changing \gamma. However, Fig. 9 shows that both short and long-term memory increase with \gamma.
3) There are some papers in the QRC literature that have already used coupled quantum oscillators or single oscillators with Kerr nonlinearity: Khan et al. ArXiv:2110.13849, Dudas et al. npj quantum info. 2023, Dudas et al. IEEE 2022, Govia et al. PRR 2021, Kalfus et al. PRR 2021. The authors could briefly explain the similarities and differences between their model and previous proposals, such as input encoding, dissipation, etc. Notice that ref. Khan et al. ArXiv:2110.13849 had already introduced the cumulant expansion to study this type of models.
4) Typos:
- The two paragraphs between the lines 41 and 61 contain the same information.
- Figures 6-9 are missing the labels.
Reviewer 2 Report
Comments and Suggestions for Authors This paper investigates a minimal model consisting of two coupled Kerr-nonlinear oscillators. The authors apply partial information decomposition(PID) to decompose information into redundancy, uniaue information and synergy. It is observed that the system transitions from predominantly redundant encoding to synergistic encoding near the critical point of dynamical instability. Additionally, the impact of high dissipation on redundancy and synergy is analyzed. Suggestions for Improvement: 1. Add more details to the calculation process: For example, in Section 2.3, the calculation of quantum mutual information could include more detailed explanations. 2. Enhance the discussion of the figures: • For Figure 4: Specifically discuss the three scenarios presented in the figure, including the states of soft modes and synergistic encoding under different conditions. • For Figure 5: Expand the discussion to include the roles of slow modes and fast modes in the system’s dynamics and the changes in synergistic effects as the coupling strength increases. 3. Include qualitative discussions (without quantitative analysis): Based on the current conclusions, consider adding discussions about how actual experimental errors and noisy conditions might influence the trends and implications of the conclusions. This would provide a broader perspective on the robustness and practical applications of the findings. Recommendation: The paper is suitable for publication after making revisions based on the points mentioned above.Author Response
Please see the attachment.

Round 2
Reviewer 1 Report
Comments and Suggestions for Authors
The Authors have responded satisfactorily to the points I made.